# Drying of Drill Cuttings: Emphasis on Energy Consumption and Thermal Analysis

**Esra Tınmaz Köse** 

Department of Environmental Engineering, Çorlu Engineering Faculty, Namık Kemal University, 59860 Çorlu/Tekirdağ, Turkey; etinmaz@nku.edu.tr; Tel.: +90-282-250-23-57

**Abstract:** Drill cuttings, contaminated with drilling fluids, are characterized by their high moisture content, which can cause problems for collection, storage, and transportation. Additionally, the practice of disposing waste with high moisture content into sanitary landfills is undesirable and mostly forbidden. For that reason, drying of waste with high moisture content, such as drill cuttings, is an essential operation. In this work, microwave and conveyor belt drying processes for drying drill cuttings containing water-based drilling fluids were examined in a lab-scale study. The results of the study indicated that the microwave dryer has been shown to be advantageous in terms of time and energy consumption for drying of thin film layers, while the conveyor drying system was more appropriate for bulk drying.

**Keywords:** drill cuttings; drying; energy; microwave; conveyor belt

## 1. Introduction

During drilling operations, it is necessary to keep the base clean by removing rock fragments and crumbs that are cut by the underground drill. During the drilling process, the structure formed by the rock fragments and crumbs cut by the drilling fluid is called interruption. Drill cuttings (DC) can have different characteristics depending on the purpose of the drilling, the characteristics and depth of the formations that are drilled through, and the characteristics of the drilling fluid used [1]. The waste produced from drilling operations consists of drill cuttings contaminated with drilling fluids [2].

The management of waste generated during oil-drilling in our country is carried out within the scope of Waste Management Regulation, which was published in the Official Gazette on 02.04.2015 and numbered as 29,314 [3]. According to regulation, drill wastes are defined by the section code "01 05—Wastes generated during the search, extraction, operation, and physical and chemical treatment of mines—drilling muds and other drilling wastes". Drilling wastes are considered as possibly dangerous waste, according to their properties and concentrations.

Drill cuttings are characterized by their high moisture content and low bulk density, which result in a low conversion efficiency as well as difficulties in its collection, storage, and transportation. The level of moisture content of drill cuttings is a critical factor that determines its disposal options [4–6].

The high moisture content of drill cuttings can cause them to have low calorific values. It is therefore not suitable for use in refused derived fuel (RDF) or direct combustion operations. Another waste-management practice is the use of drill cuttings as construction building materials. However, they are unable to be used as raw materials due to their high moisture content and irregular distribution of particle sizes. The final disposal method of drill cuttings, which are unlikely to be recovered, is sanitary landfilling. However, the regulations on the acceptance of liquid into landfill areas forms one of the most important problems in landfilling, as liquid wastes can only be taken if they are the result of specific analysis of wastewater treatment plants, or if they are re-injected into the well. The accumulation of all waste (solid–liquid) products in a single well causes the liquid waste to contain

colloidal particles. Therefore, the injection method is not an acceptable option since injection of these particles can affect production values. The temporary storage of liquid wastes in large pools is another method of waste-management. However, the long-term management of accumulated liquid waste can lead to larger problems. Therefore, taking into account the obligation of Article 9 (a) of the Waste Management Regulation [3], and Article 8 (a) of the Mining Waste Regulation [7], measures should be taken to minimize the amount of waste produced instead [8].

The drying process is an effective solution, which reduces the weight and volume of the sludge and reduces transportation and management costs, thus allowing for the easy handling, preservation, and storage of the wastes [6,9].

Drying refers to the process of thermally removing volatile substances (moisture) to yield a solid product. During the drying of wet solids, two separate processes occur simultaneously:

- Process 1—the transfer of energy (mostly as heat) from the surrounding environment to evaporate surface moisture;
- Process 2—the transfer of internal moisture to the surface of the solid and its subsequent evaporation due to Process 1 [9].

In Process 1, the removal of water from solid material as vapor occurs from the surface of the material depending on the external conditions of temperature, pressure, area of exposed surface, air humidity, and flow. In Process 2, the movement of moisture internally within the solid material depends on the physical nature, temperature, and initial moisture content of the material. While these two processes occur simultaneously during the drying process, one of the processes may be a limiting factor [9]. Drying is a complicated process that involves simultaneous heat and mass transfer, accompanied by physicochemical transformations. Based on the mechanism of heat transfer that is employed, drying is categorized into direct (convection), indirect or contact (conduction), radiant (radiation), and dielectric or microwave (radio frequency) drying. Although more than 85% of industrial dryers are of the convective type, contact dryers offer higher thermal efficiency and have economic and environmental advantages over convective dryers [10,11]. Rising energy costs, compulsory legislation on pollution, working conditions, and safety, have a direct bear upon the design as well as selection of dryers [9]. Table 1 shows the classification of dryers based on various criteria [9–12].

**Table 1.** Classification of dryers.

| Criterion | Types |
|---|---|
| Mode of operation | Batch, continuous |
| Heat input type | Convection, conduction, radiation, electromagnetic fields, combination of heat transfer modes intermittent or continuous adiabatic or non-adiabatic |
| State of material in dryer | Stationary moving agitated, dispersed |
| Operating pressure | Vacuum, atmospheric |
| Drying medium (convection) | Air, superheated steam flue gases |
| Drying temperature | Below boiling temperature, above boiling temperature, below freezing point |
| Relative motion between drying medium and solids | Co-current countercurrent mixed flow |
| Number of stages | Single multistage |
| Residence time | Short (60 min) |

In this work, an experimental study was performed on the drying of drill cuttings containing water-based drilling fluids with the two drying systems—microwave and conveyor belt dryers. There have been many studies previously conducted on the process of microwave drying drill cuttings.

In microwave drying systems, mass and treatment time directly relate to specific energy [2]. In previous literature, it was emphasized that microwave drying was a cost-effective and time-efficient system for the management of waste that was contaminated with petroleum hydrocarbons [2,6,13–20]. It also has several advantages with respect to the environment such as being a cleaner energy source, and being more energy-saving [6,21–24]. Conveyor belt dryers are versatile and suitable dryers for drying of varied products such as nuts, animal feeds, briquettes, rubbers [25], and biofuels [26]. Hot air is forced up through the product while it is carried through the dryer on conveyors [9,25]. Conveyor belt dryers are preferable drying systems for industrial applications, however, studies in the literature regarding drying of drill cuttings with conveyor belt dryers are quite limited. The aim of this study was to compare these two drying systems in respect of drying time and energy consumption. There are no studies on the drying of drill cuttings resulting from drilling operations occurring in the Thrace Region, and the material used in this study is original because of two different reasons. The first reason is that the properties of drill cuttings differ depending on the geological formation of the drilling area. The region where the drill cuttings are provided is located on five different bases starting from the Eocene in the Thrace Basin. These bases are Istranca massif and Upper Cretaceous Volcanics (Yemislicay Formation) in the North, Paleozoic sediments in the East, and the Sakarya and Intra-Pontide Suture Zones in the South. The location where the sample was collected mostly from was Istranca metamorphics [27]. Changes in geological formation at different locations, even in the same area, cause changes in the properties of drill cuttings. Properties of drilling fluid are the second reason why drill cuttings used in this study were original materials. During the preparation of the water-based drilling fluid, a large amount of water was absorbed by the clay and a suspension was formed in the drilling fluid formed by the mixture of water, clay (bentonite), and other chemicals. Some free-water circulated in this suspension. While the free-water is easily removed from drill cutting by heat, it is more difficult to remove the water absorbed by the clay [28]. This shows the difference between the materials used in this study, and the materials that are subject to many other studies in the literature.

## 2. Materials and Methods

### 2.1. Material Characterization

A sample of drill cuttings containing water-based drilling fluids was obtained from hydrocarbon drilling operations conducted in the Thrace Region of Turkey. The characterization study of drill cuttings was accomplished in the prior project of the author of [27]. Table 2 shows the parameters analyzed and the analysis methods used on them. X-ray fluorescence (XRF) data for major oxides and elements, BTEX, PCBs, mineral oil contents, total organic carbon, dissolved organic carbon, total dissolved solid and conductivity values are given in Tables 3 and 4.

**Table 2.** Parameters analyzed and analysis methods.

| Parameter | Analytical Method |
|---|---|
| DOC (Dissolved organic carbon) | SM-5310 B High-temperature combustion method |
| TDS (Total dissolved solid) | SM-2540 C gravimetric method |
| TOC (Total organic carbon) | SM-5310 B High-temperature combustion method |
| Conductivity | ASTM D1125-14 |
| BTEX | EPA-8015C Nonhalogenated organics using GC/FID |
| PCBs | ISO 10382 GC method with electron capture detection |
| Mineral oil (C10–C40) | BS EN 14039 |
| Chemical properties moisture content | X-Ray fluorescent spectrometer (XRF) ASTM 3173 |

**Table 3.** Chemical composition of drill cuttings.

| Element | % | Oxide | % | Element | % | Oxide | % |
|---------|---|-------|---|---------|---|-------|---|
| O | 41.590 | | | Ti | 0.362 | $TiO_2$ | 0.605 |
| Na Mg | 2.150 | $Na_2O_3$ | 2.898 | Cr | 0.041 | $Cr_2O_3$ | 0.060 |
| Mg | 2.482 | MgO | 4.115 | Mn | 0.084 | MnO | 0.108 |
| Al | 7.150 | $Al_2O_{33}$ | 13.509 | Fe | 3.952 | $Fe_2O_3$ | 5.650 |
| Si | 21.482 | $SiO_2$ | 45.958 | Co | 0.012 | $Co_3O_4$ | 0.016 |
| P | 0.070 | $P_2O_5$ | 0.161 | Ni | 0.017 | NiO | 0.022 |
| S | 0.506 | $SO_3$ | 1.263 | Cu | 0.042 | CuO | 0.053 |
| Cl | 3.009 | Cl | 3.009 | Zn | 0.011 | ZnO | 0.014 |
| K | 4.565 | $K_2O$ | 5.499 | Ba | 1.394 | BaO | 1.557 |
| Ca | 11.080 | CaO | 15.503 | | | | |

**Table 4.** Chemical properties of drilling cutting.

| Parameter | Unit | Value |
|-----------|------|-------|
| BTEX | mg/kg | <0.5 |
| PCBs | mg/kg | <0.1 |
| Mineral oil (C10–40) | mg/kg | 1247 |
| TOC | % | 0.7228 |
| DOC | mg/L | 209.79 |
| TDS | mg/L | 2.810 |
| Conductivity | S/cm | 4380 |
| Moisture content | % | $45 \pm 2$ |

According to the results of the measurements, the element with the highest percentage by weight of total mass was silicon (Si) with 21.482%, followed by calcium (Ca) with 11.080%. Oxide ratios of the same elements were determined as 45.958% for $SiO_2$, and 15.503% for CaO. Al and $Al_2O_3$ ratios of drill cuttings were 7.150% and 13.509% respectively. The BTEX, PCB and mineral oil concentrations were measured as 0.5 mg/kg, lower than 0.1 mg/kg and 1247 mg/kg, respectively. Initial moisture content of drill cuttings was measured as $45 \pm 2$%.

*2.2. Experimental Setup*

A programmable microwave oven (Arçelik MD 554, Turkey) with maximum output of 800 W at a frequency of 2450 MHz was used for microwave drying experiments. The scheme of the microwave oven is illustrated in Figure 1. The dimensions of the microwave inner case were 455 × 281 × 325 mm. Three different microwave output powers (120, 350, and 600 W) were taken into consideration in the drying experiments.

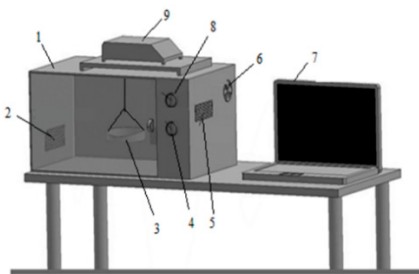

**Figure 1.** Microwave drying system: (**1**) Microwave oven, (**2**) ventilation holes, (**3**) tray, (**4**) timer, (**5**) magnetron, (**6**) fan, (**7**) computer, (**8**) power switch, (**9**) scales.

The other drying system used in the study was the conveyor dryer with dimensions of 2370 × 50 × 40 mm and with 2000 W heating power. Drying experiments were performed at 3 different

temperatures (70 °C, 80 °C, and 90 °C) and a constant band speed of 0.12 m/min. The scheme of the conveyor dryer is given in Figure 2.

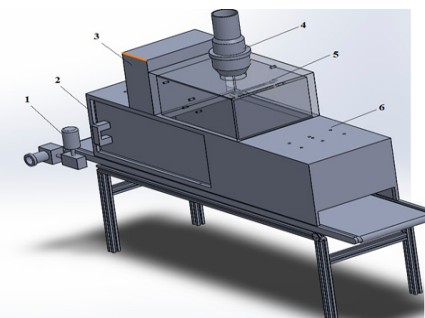

**Figure 2.** Conveyor drying system: (**1**) Electric motor, (**2**) drying room, (**3**) control panel, (**4**) fan, (**5**) heater, (**6**) ventilation hole.

Samples were taken, periodically, from each drying system in order to measure their average weight with a Presica XB 620 M (Precisa Instruments AG, Dietikon, Switzerland). The energy consumption of the microwave oven was determined using a digital electric counter with 0.01 kW h precision. Temperature change photographs of drill cuttings were taken with the thermal imager (Flir Ex E6, Estonia) before starting the experiments, and during the drying processes. Moisture content of the samples were measured within an INGDA KH-35A (China) brand oven.

*2.3. Experimental Procedure*

At the beginning of the drying experiments, the drill cuttings were homogenized by mixing. All drying tests were carried out in 50 g, 100 g, and 150 g samples. Microwave drying experiments were carried out at 120 W (at 2 min intervals), at 350 W (at 1 min intervals), and at 600 W (at 2 min intervals) microwave power levels.

Conveyor drying experiments were carried out at 70 °C (at 60 min interval), 80 °C (at 30 min interval), and 90 °C (15 min interval) at a constant belt speed of 0.12 m/min and at constant air velocity of 1 m/s.

Sample weights were measured at all powers, drying temperatures in both drying systems, and thermal images of the samples were taken. The thermal images showed the regions where the samples were most heated and whether they were homogeneous or not.

The moisture content on the wet base was defined as the ratio of water weight in the sample to the total weight of the sample. Equation (1) was used to calculate the moisture content on wet base [29,30].

$$ m = \frac{M_w}{M_w + M_s} \tag{1} $$

where $M_w$ (g) was the weight of water in the sample and $M_s$ (g) was the weight of the dry sample.

Energy consumption was measured and recorded to determine the energy consumption of each dryer by means of the energy measurement device at the measurement periods. The studies were done with three replicates at all power, and drying temperatures.

## 3. Results and Discussion

*3.1. Drying Characteristics*

Drying experiments were completed when the drill cuttings were completely dried. Drying processes were applied until the weight of the sample reduced to a level corresponding to moisture content of approximately 13.5 ± 0.5% on a wet basis for microwave drying, and approximately 13.5 ± 0.3% on wet basis for conveyor belt drying, while the initial moisture content of drill cuttings were

approximately 42 ± 2%. The variation in the moisture content of drill cuttings for different weights in respect to the wet base are given in Figure 3 for the microwave dryer, and Figure 4 for the conveyor belt dryer.

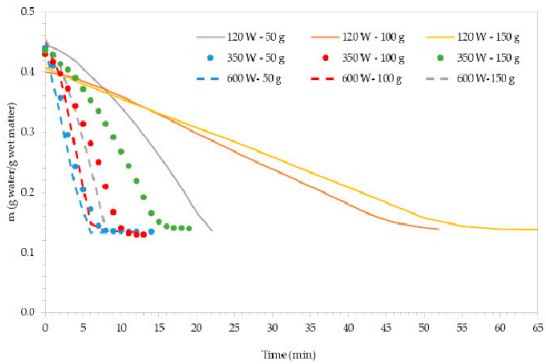

**Figure 3.** Variation of moisture content of drilling cutting for microwave dryer.

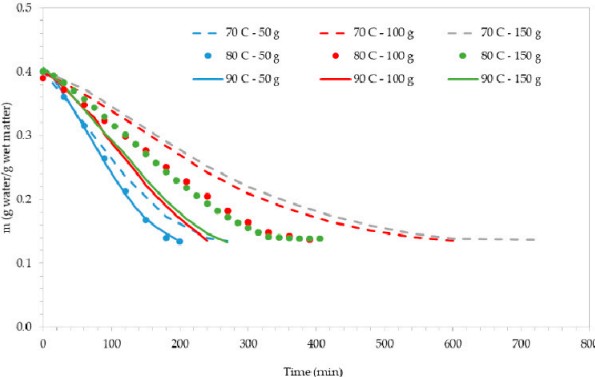

**Figure 4.** Variation of moisture content of drilling cutting for conveyor belt dryer.

According to results, it was clear that the drying time in the microwave dryer was influenced by microwave power. Moisture loss accelerated and drying time was shortened with increased microwave power. The drying times of the microwave dryer for 120, 350, and 600 W microwave powers were; 22, 14, and 8 min, respectively, for 50 g samples; 52, 13, and 12 min for 100 g samples, and 65, 19, and 8 min for 150 g samples. Past studies have showed that drying times for microwave drying of hydrocarbon drilling sludge decreased with increasing microwave power levels, whereas drying time increased with increasing layer thickness of the sample [6,31]. Tınmaz Köse et al. [6] reported that drying times were found as 8.5, 2.5, and 2 s for 50 g samples; 37.5, 6.5, and 5 s for 150 g samples at 120, 460, and 600 W, respectively. Tınmaz Köse et al. [31] concluded that when microwave power level was increased from 120 W to 700 W, the drying time decreased from 48 min to 5.5 min. The results obtained in this study were consistent with data from previous literature.

For conveyor belt dryers, it was determined that drying temperature was effective on the moisture content of drill cuttings and drying times were decreased with increasing drying temperature. The drying times of conveyor dryers at 70 °C, 80 °C, and 90 °C temperatures were 270, 200, and 195 min for 50 g samples, respectively; 240, 390, and 600 min for 100 samples, and 270, 405, and 720 min for 150 g samples.

The working mechanisms of the two dryers were different. In microwave applications, microwaves can transport energy to the entirety of the material, as energy affects the internal structure. This is the main reason why microwave drying systems shorten drying times.

*3.2. Energy Consumption*

Energy consumptions of the microwave dryer at different microwave power levels are given in Figure 5, and energy consumptions of the conveyor dryer at different drying temperatures are shown in Figure 6. During the drying period, energy consumptions were measured and recorded with an energy meter for both drying systems. Energy consumptions of the microwave dryer for 120, 350, and 600 W microwave powers were 0.33, 0.13, and 0.13 kWh for 50 g samples, respectively; 0.78, 0.13, and 0.18 kWh for 100 g samples, and 0.65, 0.19, and 0.12 kWh for 150 g samples. Energy consumptions of the conveyor dryer to reach the drying temperatures of 70 °C, 80 °C and 90 °C were 0.45 kWh, 0.95 kWh, and 1.21 kWh, respectively. At the end of the drying process, energy consumptions of the conveyor dryer at 70 °C, 80 °C and 90 °C temperatures were 6.80, 4.08, and 7.02 kWh minutes for 50 g samples, respectively; 14.65, 6.76, and 8.35 kWh for 100 samples, and 16.75, 13.59, and 10.41 kWh for 150 g samples.

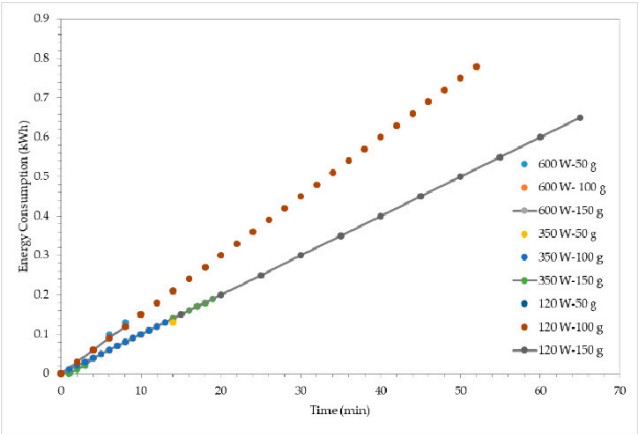

**Figure 5.** Energy consumption of the microwave dryer at different microwave power levels.

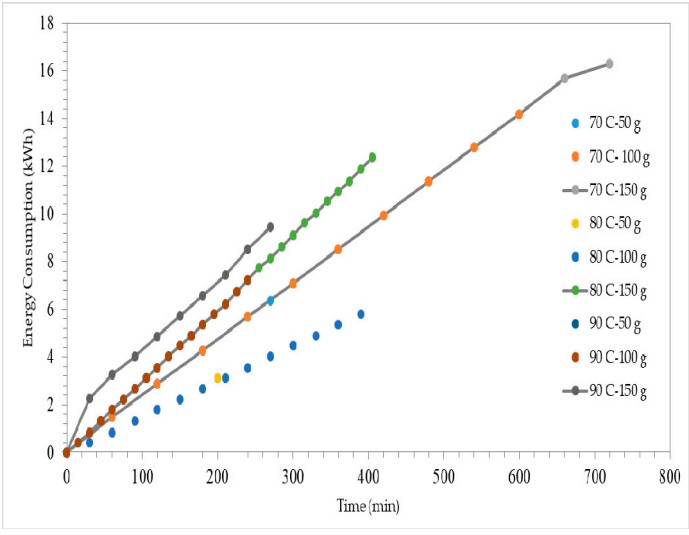

**Figure 6.** Energy consumption of the conveyor dryer at different drying temperatures.

Previous studies [6,31] reported that energy consumption values for microwave drying of hydrocarbon drilling sludge decreased with increasing microwave power levels because of decreasing drying time. Tınmaz Köse et al. [6] found that energy consumptions values for microwave power levels of 120 W, 460 W, and 600 W were recorded as between 0.03 and 0.11 kWh, 0.02 and 0.08 kWh, and 0.01 and 0.07 kWh, respectively. Tınmaz Köse et al. [31] determined that energy consumptions for

microwave power levels of 700 and 120 W were 0.11 and 0.16 kW/h, respectively. The results obtained in the study were consistent with previous literature.

Conveyor drying systems are open systems while microwave drying systems are closed systems. Heat losses in closed systems are less than heat losses in open systems. While open systems such as conveyor systems have insulated systems to prevent heat loss, openings of input and output structures causes heat loss. The heaters must operate continuously to prevent the loss of ambient temperature with heat loss. This situation causes the energy consumption of the conveyor drying systems to be high.

*3.3. Thermal Analysis*

Images and thermal images of drill cuttings at the beginning of the study (wet state at room temperature), during the drying process, and at the end of the study (dry state) were taken and recorded. A scale showing the minimum and maximum temperature values of photographs and thermal images are given in S1–S6. At the end of the experiments, the temperatures of dried samples were measured between 22.1 °C and 194 °C for the microwave dryer, and between 22.3 °C and 84.8 °C for the conveyor dryer.

High energy absorption and high drying rates led to local overheating during the drying process. Overheating caused excessive localization and made control difficult [31]. Increasing microwave power levels resulted with an increased temperature of the final product. In some areas the color appeared as yellow. These areas were where the product continued to warm up. Microwave energy acted on water molecules in the product and caused heat to be released by vibrating water molecules [32]. As a result of this, increased temperature was observed in the product, especially in the center. The red color on the sides indicated that the liquid part of the slurry flowed sideways, and that there was an excess of energy caused by this [32].

According to thermal images taken during the study, the drying process occurred homogenously. Generally, the same temperature values were observed on the surface of the drill cuttings, as the temperatures in the conveyor drying process acted on the surface. The observed temperature differences were related to the fact that the homogeneous structure of drill cuttings was not fully achieved.

## 4. Conclusions

Drying of drill cuttings is an effective waste management step for the reduction in transportation and management costs, supplying needs of easy handling, preservation, and storage of such waste. However, drying time and energy consumption are limiting factors in decisions surrounding suitable drying technology.

While microwave drying systems have great advantages in terms of energy consumption for thin layer drying processes, they are not suitable for drying bulk materials. When drying bulk materials, it is difficult to penetrate microwave energy into bulk materials and, therefore, drying will not occur in the internal side of the materials. In this study, the drying of samples of 50, 100, and 150 g weights, with a layer thickness of 1 cm, was determined according to the penetration depth of microwave energy, as was performed. As a result of the operating mechanism of the microwave drying processes, the drying process was carried out quickly due to the homogeneous distribution of energy in thin layer drying, and thus, the moisture was quickly removed from the material. The temperature changes of material during the drying period were proven by thermal images.

During drilling operations, a considerable amount of drill cuttings occurred. Thin layer drying may not be an applicable and economic practice for excess amounts of drill cuttings. For that reason, when considering which drying method to apply, the amount of drill cuttings should be considered.

In this study, according to results of the experiments carried out to determine the most suitable drying process, it was concluded that it would be more appropriate to prefer conveyor dryers in case of bulk drying, while providing the appropriate results for thin film layer drying by microwave dryers.

**Funding:** This research received no external funding.

**Conflicts of Interest:** The author declares no conflict of interest.

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
