# Peer review of "Drying of Drill Cuttings: Emphasis on Energy Consumption and Thermal Analysis"

_processes, doi:10.3390/pr7030145_

Round 1

Reviewer 1 Report

The author should call the type of material to be drilled, both in the abstract and introduction.

Citing the Turkish regulation in the abstract is technically irrelevant

figure showing the studied process.

ARTICLE SIMILAR TO:

https://onlinelibrary.wiley.com/doi/full/10.1002/ep.13104

please explain the differences

Author Response

Response to Reviewer 1 Comments

Dear Reviewer,

First of all, I want to thank you for your kind suggestions and comments for my article.

I took your suggestions into consideration and did the corrections in the article. English language and style were checked. The corrections are given below.

Best regards.

Comment 1: The author should call the type of material to be drilled, both in the abstract and introduction.

Response 1: The material used in this study was drilling cutting containing water-based drilling fluids was obtained from hydrocarbon drilling operations conducted in the Thrace Region of Turkey. According to comment and suggestion of the reviewer the type of material was explained in abstract and also introduction.

The sentence in the submitted manuscript was “In this work, microwave and conveyor belt drying processes were examined in a lab-scale study.”.  This sentence was rearranged (in line 13-15) as given below:  

“In this work, microwave and conveyor belt drying processes for drying of drilling cutting containing water-based drilling fluids were examined in a lab-scale study.”

The sentence in the submitted manuscript was “In this work, experimental study was performed for drying of drilling cutting containing water-based drilling fluids with two drying system which were microwave and conveyor belt dryers.” This sentence was rearranged (in line 77-78) as given below: 

“In this work, experimental study was performed for drying of drilling cutting containing water-based drilling fluids with two drying system which were microwave and conveyor belt dryers.”

Comment 2: Citing the Turkish regulation in the abstract is technically irrelevant.

Response 2: The sentence in the submitted manuscript was “Additionally, disposal of highly moisture wastes in sanitary landfills are forbidden according to Turkish regulations.”.  This sentence was rearranged (in line 9-12) as given below: 

“Additionally, disposal of highly moisture wastes in sanitary landfills is undesirable and mostly forbidden practice.”

Comment 3: figure showing the studied process.

Response 3: This comment is not understood.

Comment 4: ARTICLE SIMILAR TO: https://onlinelibrary.wiley.com/doi/full/10.1002/ep.13104

please explain the differences

Response 4: This article is in the list of references:

6. Tınmaz Köse, E.; Çelen, S.; Çelik, S.Ö. Conventional and microwave drying of hydrocarbon cutting sludge, Environ Prog Sustain Energy, 2018, Article DOI: 10.1002/ep.13104.

The differences are given in Table 1.

Table 1. Differences between submitted anf referenced articles

Submitted   article

Referenced   article

Material

Drilling   cutting form Thrace Region

Cutting   sludge from Batman

Drying   processes

Microwave   and conveyor dryers

Microwave   and conventional dryers

Microwave   power level

120,   350 and 600W

120,   460 and 600 W

Examined   topics

Drying   time

Energy   consumption

Thermal   analysis

Energy   consumption

Modelling   of the drying behaviours

Reviewer 2 Report

This paper presents a study on drying of drilling cutting. The microwave and conveyor belt dryers are compared.

1.       The novelty of this study is not clear. The microwave and conveyor belt drying have been both extensively studied for various materials. What are the major differences between this study and the previous ones, except for using them to dry drilling cutting materials? What is the main new knowledge obtained by this study? The results for drying characteristics, energy consumption, and thermal analysis in this study are all similar to other previous studies. It is not clear what is the new findings here.

2.       The description of eqn. (1) is not correct. What is the definition of Mw? If it is “weight of wet sample” as described the in manuscript, then the equation seems to be incorrect.

3.       Eqn. (2) and (3) are both used to calculate dimensionless moisture ratio mr. However, there is no results or discussion about mr in this paper. These two equations are only consistent when assuming zero equilibrium moisture content, which is the case for microwave drying. However, for conveyor belt drying, these two equations will not be consistent if the equilibrium moisture content is not zero.

4.       In Fig. 3, why are the initial moisture ratios of the samples for 120W/100g and 120W/150g different from the other samples?

5.       Line 209, it says “The biggest disadvantage of the microwave drying system, which has a great advantage in terms of energy consumption, is that it is suitable for thin layer drying”. Why is being suitable for thin layer drying a disadvantage for microwave drying system?

Author Response

Response to Reviewer 2 Comments

Dear Reviewer,

First of all, I want to thank you for your kind suggestions and comments for my article.

I took your suggestions into consideration and did the corrections in the article. English language and style were checked. The corrections are given below.

Best regards.

Comment 1:       The novelty of this study is not clear. The microwave and conveyor belt drying have been both extensively studied for various materials. What are the major differences between this study and the previous ones, except for using them to dry drilling cutting materials? What is the main new knowledge obtained by this study? The results for dryharacteristics, energy consumption, and thermal analysis in this study are all similar to other previous studies. It is not clear what is the new findings here.

Response 1: The material used in this study is an original one and characteristics of its differs depending on the geological formation of the drilling area. There are no studies on the drying of wastes resulting from drilling operations in Thrace Region. This study includes preliminary data of a study carried out in order to design a pilot scale and/or real-scale dryer. Therefore, the change in moisture content during drying, energy consumption and thermal change of the product are very important data. The sentences shown in the line 87-88 were added in the article to explain that.

Comment 2:      The description of eqn. (1) is not correct. What is the definition of Mw? If it is “weight of wet sample” as described the in manuscript, then the equation seems to be incorrect.

Response 2: The abbreviations for Eq-1 were expressed wrongly. Mw shows the weight of water into sample. The explanation of the Eq-1 was rearranged (line 145) given as below.

Equation-1 was used to calculate the moisture content on wet base [28].

                                             (Eq-1);

where Mw (g) was the weight of water into sample and Ms (g) was the weight of dry sample.

Comment 3:      Eqn. (2) and (3) are both used to calculate dimensionless moisture ratio mr. However, there is no results or discussion about mr in this paper. These two equations are only consistent when assuming zero equilibrium moisture content, which is the case for microwave drying. However, for conveyor belt drying, these two equations will not be consistent if the equilibrium moisture content is not zero.

Response 3: You are right in your review. Although Eq 2 and 3 were given in the article there was no results or discussion in the paper. For that reason these two equations were removed from the article.

Comment 4.       In Fig. 3, why are the initial moisture ratios of the samples for 120W/100g and 120W/150g different from the other samples?

Response 4: There is no error in Figure 3. In fact, the initial moisture values in almost all experiments are different. A single microwave dryer was used in this study. Thus, the time difference between the use of the same dryer in different powers resulted in differences in the initial moisture content. Although the sample was stored in the desiccator, different moisture content values were measured. In addition, the initial moisture content of the sample measured at 42±2% is another reason for this difference.

Comment 5.       Line 209, it says “The biggest disadvantage of the microwave drying system, which has a great advantage in terms of energy consumption, is that it is suitable for thin layer drying”. Why is being suitable for thin layer drying a disadvantage for microwave drying system?

Response 5: This statement applies only to the material used in the study. These sentences were removed from line 209 and the explanation in the conclusion section was enlarged as given below:

“While microwave drying system has a great advantage in terms of energy consumption for thin layer drying, it is not suitable for drying of bulk material. During the drilling operations, considerable amount of drilling cutting is occurred and thin layer drying may be not applicable and economic practice for excess amount of drilling cutting. For that reason, when considering which drying method to be applied, the amount of drilling cutting should be considered.”

Round 2

Reviewer 2 Report

Most of the issues have been addressed. However, the contribution of this study is till not clear. The author mentioned that the novelty is to study the drying of wastes resulting from drilling operations in Thrace Region, which is not studied before. What is the uniqueness of the waste resulting from drilling cutting? What is the fundamental new findings for this material compared with many other materials studied before? Without describing these information, simply checking a new material cannot be considered as a significant novelty.

Author Response

Response to Reviewer 2 Comments-2nd Round

Dear Reviewer,

First of all, I want to thank you for your kind suggestions and comments for my article.

I took your suggestions into consideration and did the corrections in the article. The corrections are given below.

Best regards.

Point 1: Most of the issues have been addressed. However, the contribution of this study is till not clear. The author mentioned that the novelty is to study the drying of wastes resulting from drilling operations in Thrace Region, which is not studied before. What is the uniqueness of the waste resulting from drilling cutting? What is the fundamental new findings for this material compared with many other materials studied before? Without describing these information, simply checking a new material cannot be considered as a significant novelty. 

Response 1: The paragraph given below was added at the end of introduction section.

“….There are no studies on the drying of drilling cuttings resulting from drilling operations in Thrace Region and the material used in this study is an original material because of two different reasons. The first reason is that the properties of drilling cutting differs depending on the geological formation of the drilling area. The region where the drilling cutting is provided is located on five different bases starting from the Eocene in the Thrace Basin. These bases are Istranca massif and Upper Cretaceous Volcanics (Yemislicay Formation) in the north, Paleozoic sediments in the east, and Sakarya Zone and Intra-Pontide Suture Zones in the south. The location where the sample collected is mostly Istranca metamorphics [27]. Changes in geological formation at different locations, even in the same area, cause changes in the properties of the drilling cutting. Properties of drilling fluid is the second reason why drilling cutting used in this study is original one. During the preparation of the water-based drilling mud, a large amount of water is absorbed by the clay and a suspension is formed in the drilling mud formed by the mixture of water, clay (bentonite) and other chemicals. Some water -free water- circulates in this suspension. While the free water is easily removed from drilling cutting by heat, it is more difficult to remove the water absorbed by the clay [28]. This shows the difference between the materials used in this study and the materials that are subject to many others studies in the literature.”